# Interpretability Driven Evolutionary Approach for the Design of Biological Sequences

Akash Pandey [1]   Wei Chen [1]   Sinan Keten [1 2]

## Abstract

Designing biological sequences such as proteins and DNA for desired properties is challenging due to vast search spaces and limited wet lab evaluation budgets. Current evolutionary approaches ignore sequential dependencies and rely on random mutations, which scale poorly for long sequences. In contrast, reinforcement learning (RL) and generative models that explicitly model sequence structure, require large datasets to guide generation toward the target properties. These limitations suggest the need for a method that combines the sample efficiency of evolutionary approaches with the ability to exploit sequential structure. In this work, we propose a novel evolutionary approach, IDEAS, in which mutations are guided by an explainable model. The model identifies critical motifs in high-fitness sequences and uses them to mutate non-critical positions. Across eight continuous-property datasets, seven baselines, and three evaluation budgets, IDEAS achieves a **19%** acceleration in design while maintaining a favorable position on the Pareto curve balancing acceleration, diversity, and novelty.

## 1. Introduction

Biological sequences such as proteins and DNA play a central role in therapeutics and biotechnology, making the ability to design sequences with desired properties highly valuable (Zimmer, 2002; Lorenz et al., 2011; Barrera et al., 2016a; Ogden et al., 2019). However, the combinatorially large search space of biological sequences renders this design problem inherently challenging. Prior work has approached biological sequence design using evolutionary methods (Sinai et al., 2020; Hansen, 2006; Arnold, 1998), reinforcement learning (RL) (Angermueller et al., 2019; Jain et al., 2022), and generative model–based (Brookes et al., 2019; Brookes & Listgarten, 2018; Gupta & Zou, 2019a) approaches.

Evolutionary approaches (Sinai et al., 2020; Hansen, 2006; Bloom & Arnold, 2009) are among the most widely used methods for biological sequence design, iteratively improving candidate sequences through mutation and oracle-based selection using wet-lab experiments or computational simulations. Their key advantage lies in minimal assumptions: they require no training data and impose no constraints on the oracle, making them well-suited for black-box optimization. However, mutations are typically applied randomly without accounting for sequential dependencies, causing poor scalability with sequence length and a high demand for oracle evaluations in large search spaces.

Reinforcement learning (RL)–based methods, such as Proximal Policy Optimization (PPO) (Schulman et al., 2017) and Generative Flow Networks (GFlowNets) (Bengio et al., 2021), have been applied to biological sequence design. DynaPPO (Angermueller et al., 2019), an on-policy PPO-based method, trains solely on samples generated by the current policy, which limits exploration and restricts search-space coverage. In contrast, GFN-AL (Jain et al., 2022) is an off-policy GFlowNet-based approach that leverages sequences from previous iterations to enable the generation of high-performing, diverse, and novel designs. However, this increased novelty can produce out-of-distribution sequences, leading to unreliable reward estimates and potentially degrading policy learning (Trabucco et al., 2021; Yu et al., 2021).

Another class of design methods is based on generative models, including generative adversarial networks (GANs) (Goodfellow et al., 2014) and variational autoencoders (VAEs) (Kingma & Welling, 2019), which are conditioned to produce sequences with desirable properties. Design by adaptive sampling (Dbas) (Brookes & Listgarten, 2018) iteratively shifts the generative distribution towards desired sequences by weighting samples according to their oracle-evaluated property, while Conditioning by Adaptive Sampling (Cbas) (Brookes et al., 2019) further incorporates a

---

[1]Department of Mechanical Engineering, Northwestern University, Evanston, USA [2]Department of Civil and Environmental Engineering, Northwestern University, Evanston, USA. Correspondence to: Sinan Keten <s-keten@northwestern.edu>.

*Proceedings of the 43rd International Conference on Machine Learning*, Seoul, South Korea. PMLR 306, 2026. Copyright 2026 by the author(s).

penalty based on divergence from the initial data distribution to mitigate pathologies due to out-of-distribution sequences. Although these approaches can generate novel sequences, they typically require a large number of oracle evaluations (often exceeding $10^3$) to reliably move the distribution towards the desired region, limiting their applicability under tight evaluation budgets (Brookes & Listgarten, 2018; Brookes et al., 2019; Gupta & Zou, 2019a).

Existing biological sequence design methods either fail to scale to long sequences due to random mutations that ignore sequential dependencies, or require a large number of oracle evaluations (often exceeding $10^2$) to learn distributions over high-performing sequences. In practical experimental and computational settings, only a limited number of oracle evaluations can be performed per design cycle, making such requirements prohibitive (Graham et al., 2025; Tokareva et al., 2014). Consequently, there remains a critical gap for methods that combine the sample efficiency of evolutionary approaches with the ability of RL- and generative models to capture sequential dependencies. Bridging this gap is essential for developing design frameworks that are both sample-efficient and effective under tightly constrained oracle budgets.

**Our Contribution** In this work, we introduce IDEAS (**I**nterpretable **E**volutionary **A**pproach for biological **S**equences), which combines explainable models (XAI) that capture sequential dependencies with evolutionary design strategies. By leveraging attribution scores from XAI, IDEAS identifies the critical contiguous regions of top-performing sequences that drive the target property, as well as non-salient regions. This information guides *interpretable mutations*, where non-salient regions are selectively replaced or modified using motifs from critical regions, enabling more informative exploration of the sequence space compared to random mutations. Consequently, IDEAS accelerates sequence optimization while maintaining biological plausibility and interpretability.

The key contributions of this work are threefold: (1) we propose IDEAS, an interpretable evolutionary framework that leverages explainable models (XAI) to guide mutations and accelerate exploration of the biological sequence design space; (2) we conduct a comprehensive empirical evaluation of IDEAS against seven state-of-the-art baselines across eight biological design tasks and multiple oracle evaluation budgets, using design acceleration, diversity, and novelty as evaluation metrics; and (3) through extensive ablation studies, we analyze the impact of different mutation strategies within IDEAS and demonstrate that a wide range of faithful XAI methods can be seamlessly integrated as plug-and-play components.

## 2. Problem Formulation

In this work, given an initial dataset $\mathcal{D}_0 = \{(\mathbf{x}_i, y_i)\}_{i=0}^{N_0-1}$, where $y_i$ denotes the continuous property value of sequence $\mathbf{x}_i$, we aim to design biological sequences $\mathbf{x} \in \mathcal{V}^L$ that exhibit desired properties. Here, $\mathcal{V}$ denotes the vocabulary (e.g., amino acids or nucleotides), and $L$ denotes the sequence length. Designed sequences are evaluated using a black-box oracle model $f : \mathcal{V}^L \to \mathbb{R}$. These oracle models are accurate but expensive, such as wet-lab experiments or high-fidelity computational simulations.

Given the vast search space of biological sequences ($|\mathcal{V}|^L$), exhaustive oracle evaluation is infeasible. However, recent advances in experimental techniques and their reduced turnaround times have made active learning a practical approach for iterative design with feedback at each round. In this work, we perform ten rounds of iterative design, querying $B$ new sequences per iteration to discover sequences with high $f(\mathbf{x})$, diversity, and novelty. In practice, the number of allowable oracle evaluations is often limited. To reflect this constraint, we compare different design methods under oracle budgets of $B \in \{20, 50, 100\}$.

## 3. IDEAS: Interpretability Driven Evolutionary Approach for Biological Sequences

To address the poor scalability of traditional evolutionary methods to long sequences and the large data requirements of generative model–based approaches, we propose a novel active learning framework for biological sequence design. Our method leverages an explainable model (XAI) to quantify position-level attribution scores, which are then used to guide interpretable and informed mutations, enabling sample-efficient exploration of the design space under constrained oracle budgets. Even with the integration of XAI into the mutation-based design process, the wall-clock time remains lower than that of generative and reinforcement learning–based methods, as shown in Appendix C.3.

**Definition 3.1** (Position-wise attribution scores)**.** Given a biological sequence $x_i \in \mathcal{V}^L$ with property $y_i = f(x_i)$, a position-wise attribution score $\phi_i \in \mathbb{R}^L$ is defined as qualitative contribution of each position $j \in [0, L-1]$ in $x_i$ towards $y_i$.

Our explainability-driven design method, IDEAS, consists of four distinct steps. Since the design process proceeds iteratively, we describe the operations performed at the $t$-th iteration in detail below.

**Step 1 (Selecting top sequences)**: The top $B$ sequences from $\mathcal{D}_{t-1}$ (dataset at $t-1$ iteration) is selected as:

$$\mathcal{D}^{\text{top}} = \{(\mathbf{x}_i, y_i \in \mathcal{D}_{t-1} : y_i \text{ ranks in top } B\} \quad (1)$$

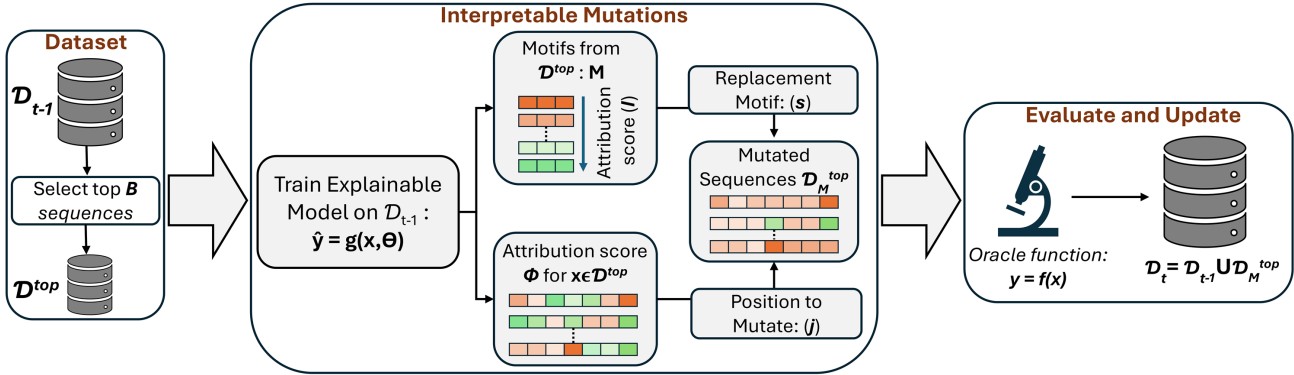

*Figure 1.* The active learning process of IDEAS. At iteration $t$, the top-$B$ sequences from $\mathcal{D}_{t-1}$ form $\mathcal{D}^{\text{top}}$. An explainable model $g(\theta)$ trained on $\mathcal{D}_{t-1}$ extracts motifs $\mathbf{M}$ and position-wise attribution scores $\phi_i$ for sequences in $\mathcal{D}^{\text{top}}$. Contiguous positions with low $\phi_i$ (green) are probabilistically selected for mutation, while high-attribution motifs $I$ (red) are selected as replacements. The resulting mutated sequences $\mathcal{D}^{\text{top}}_M$ are evaluated by the oracle $f$ and added to $\mathcal{D}_{t-1}$ to form $\mathcal{D}_t$.

**Step 2 (Training XAI model)**: Using $\mathcal{D}_{t-1}$, an XAI model ($g$) is trained such that

$$\hat{y}_i = g(x_i; \theta) \qquad (2)$$

Once the $\theta$ is optimized, XAI can quantify the attribution score of each position in the sequence as

$$\phi_i = g_{att}(x_i, \theta^*, \hat{y}_i), \text{ where, } \phi_i \in \mathbb{R}^L \qquad (3)$$

**Step 3 (Extracting Motifs)**: In this study, we define motif $s_j$ as the contiguous sub-segment of length $m$ within a biological sequence such that $s_j \in \mathcal{V}^m$. The attribution score, $I_j$, of $s_j$ in $x_i$ is calculated using $\phi_i$ in Equation. 3 as:

$$I_j = \frac{1}{m} \sum_{k=j}^{j+m-1} \phi_i[k], \ j \in [0, L-m] \qquad (4)$$

The motif size $m$ in Equation 4 is a tunable hyperparameter that can be selected based on domain expertise. In this work, we set $m = 1$ for all datasets except for the two FLIP datasets (Dallago et al., 2021), for which $m = 10$ is used (see Appendix C.4). As shown in Appendix C.2, increasing $m$ can improve design performance, but the gains plateau rapidly. All $s_j$'s and their corresponding $I_j$'s extracted from $x_i$ are stored in a list as $\mathcal{M}_i = \{(s_j, I_j) : j \in [0, L-m]\}$. Similarly, motifs and their corresponding attribution scores are extracted from all top $B$ sequences in $\mathcal{D}^{\text{top}}$ and aggregated as follows:

$$\mathbf{M} = \begin{bmatrix} \mathcal{M}_0 | \mathcal{M}_1 | \dots | \mathcal{M}_{B-1} \end{bmatrix}, \ |\mathbf{M}| = n_m \qquad (5)$$

It is important to note that the repeating motifs $s_j$ in $\mathbf{M}$ are collapsed into one entry by averaging their corresponding $I_j$ values.

**Step 4 (Interpretable Mutations)**: In this step, each sequence in $\mathcal{D}^{top}$ is mutated with motifs from $\mathbf{M}$ to maximize

$y$. For mutations, firstly, the attribution scores $\phi_i$ at the position/monomer level are converted to the motif level using

$$\psi_i[j] = \sum_{k=j}^{j+m-1} \phi_i[k], \ j \in [0, L-m] \qquad (6)$$

such that $\psi_i \in \mathbb{R}^{L-m+1}$. The scores $\psi_i[j]$ are then transformed into a probability distribution as follows:

$$p_i[j] = \frac{\exp(-\psi_i[j]/\tau_1)}{\sum_{k=0}^{L-m} \exp(-\psi_i[k]/\tau_1)}, \quad j = 0, \dots, L-m, \qquad (7)$$

where the higher $p_i[j]$ values are assigned to motifs with smaller attribution score. The temperature parameter $\tau_1$ controls the concentration of the probability distribution: a smaller $\tau_1$ amplifies differences in $\psi_i$ values, leading to sharper probabilities that favor positions with low importance scores, while a larger $\tau_1$ leads to more uniform assignment of probability scores. Similarly, we convert the motif attribution scores in $\mathbf{M}$ into probabilities using a temperature parameter $\tau_2$:

$$p_f[s] = \frac{\exp(I_s/\tau_2)}{\sum_{k=0}^{n_m-1} \exp(I_k/\tau_2)}, \quad s = 0, \dots, n_m - 1, \qquad (8)$$

where $I_s$ denotes the attribution score of the $s$-th motif in $\mathbf{M}$. Higher $p_f[s]$ values correspond to motifs with larger attribution scores. For each mutation in $x_i$, we select: (i) a starting position $j$ in sequence $\mathbf{x}_i$ to mutate, and (ii) the index $s$ of replacement motif from $\mathbf{M}$, as follows:

$$j \sim \text{Categorical}(p_i[0], p_i[1], \dots, p_i[L-m]) \qquad (9a)$$
$$\mathbf{s} \sim \text{Categorical}(p_f[0], p_f[1], \dots, p_f[n_m-1]) \qquad (9b)$$

Using the above equations, we perform **two mutations** for each sequence $\mathbf{x}_i \in \mathcal{D}^{\text{top}}$: (i) an **exploitative mutation** with $\tau_1 = \tau_2 = 1$, and (ii) an **exploratory mutation** with

$\tau_1 = \tau_2 = 1000$. The resulting mutated sequence is denoted by $\mathbf{x}_i^M$. Each mutated sequence is evaluated using the oracle and collected into $\mathcal{D}_M^{\text{top}} = \left\{(\mathbf{x}_i^M, f(\mathbf{x}_i^M))\right\}_{i=0}^{B-1}$, which satisfies $|\mathcal{D}_M^{\text{top}}| = B$. The dataset for the next iteration is then updated as $\mathcal{D}_t = \mathcal{D}_{t-1} \cup \mathcal{D}_M^{\text{top}}$. One iteration ($t$) of the design process is illustrated in Figure 1.

The mutation strategy described between Equations 4–9 represents just one possible approach. Depending on the temperature parameters $\tau_1$ and $\tau_2$, IDEAS can implement a variety of mutation strategies. Through an ablation study, we empirically demonstrate that the current strategy yields the most effective design performance.

### 3.1. XAI Model

As shown in Equations 2 and 3, an explainable model (XAI) is required to compute attribution scores $\phi_i$. Two classes of methods can provide these scores: (i) post-hoc explainability techniques such as DeepLift (Shrikumar et al., 2017), Integrated Gradients (Sundararajan et al., 2017), or GradientSHAP (Lundberg & Lee, 2017), and (ii) architecturally interpretable models such as attention-based Transformers (Vaswani et al., 2017; Wu et al., 2020). Any method that produces faithful (i.e., reality-consistent) attribution scores (Dasgupta et al., 2022; Pandey et al., 2025) is compatible with the IDEAS framework. As shown empirically in Section 5.3.3, the choice of XAI model does not affect design performance as long as the resulting attribution scores are comparably faithful.

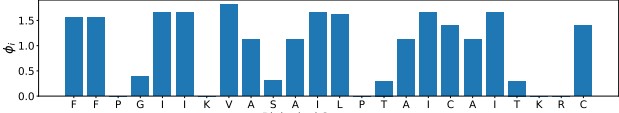

*Figure 2.* Position-wise attribution scores $\phi_i$ for a sample sequence $x_i$ from the GRAVY index dataset (Appendix B).

Based on the work of Pandey et al. (2025) on the XAI model, we adopt COLOR as our XAI. COLOR segments input sequences $x_i$ of length $L$ into $L - m + 1$ overlapping motifs of size $m$, learns their latent representations, and models motif interactions through linear transformations in latent space. This architecture enables efficient computation of position-wise attribution scores $\phi_i$. We select COLOR for two key advantages: (i) faithful (aligns with the ground-truth) attribution scores, and (ii) less trainable parameters that ensure sample-efficient learning from limited data. The attribution scores from COLOR in its original form capture the magnitude of impact but not directionality (i.e., whether a position in $x_i$ positively or negatively influences $y$). We modify COLOR to incorporate directional attribution, with details provided in Appendix A.

In Figure 2, we visualize the attribution scores $\phi_i$ for a representative sample $x_i$ from the GRAVY index dataset, which we discuss in detail later. The figure shows that

amino acids V, I, and F receive the highest attribution scores, consistent with the analytical form of the GRAVY index provided in Appendix B. This agreement demonstrates that the COLOR method produces faithful attribution scores.

### 3.2. Metrics to quantify the performance

We compare sequence design methods based on their ability to generate better, diverse, and novel sequences (Bengio et al., 2021). While we adopt the diversity and novelty definitions from Bengio et al. (2021), we reformulate the quality metric to emphasize sample efficiency by measuring which method generates high-property sequences with fewer oracle evaluations. We quantify this through Area Under the Curve (AUC), computed as:

$$\text{AUC} = \int_0^T \frac{1}{B} \sum_{\mathbf{x} \in \mathcal{D}^{\text{top}}(t)} f(\mathbf{x})\, dt, \tag{10}$$

where $T$ is the number of design iterations.

## 4. Related Work

Deep learning models have been extensively combined with post-hoc XAI methods, such as DeepLIFT and Integrated Gradients, to identify critical motifs in biological sequences, particularly DNA. These approaches have been successfully applied to several tasks, including binding affinity prediction, where attribution scores from XAI methods reveal sequence regions that drive model predictions (De Almeida et al., 2022; Shrikumar et al., 2018; Avsec et al., 2021; Horton et al., 2023; Seitz et al., 2024). Importantly, several of these studies provide experimental validation, demonstrating that XAI methods can reliably distinguish functionally critical positions (or motifs) from non-salient positions in biological sequences (De Almeida et al., 2022).

More recently, Seitz et al. (2025) leveraged explainable models in conjunction with a cluster summary matrix (CSM) to systematically identify positions essential (non-mutable) for DNA functionality, as well as positions where mutations most strongly affect functional outcomes. This line of work further reinforces the utility of attribution-based explanations for uncovering mechanistic insights into sequence-function relationships.

In parallel, the advent of Transformer-based architectures (Vaswani et al., 2017) has enabled powerful sequence-to-property prediction models for biological data. Several studies have investigated whether attention weights in these models inherently correspond to biologically meaningful or functionally important positions in sequences (Vig et al., 2020; Liu et al., 2024; Karimi et al., 2020). However, recent evidence suggests that attention scores are often poorly aligned with ground-truth attribution and can be unreliable as faithful explanations of model behavior in biological sequences (Pandey et al., 2025).

*Table 1.* Area under the curve (AUC) across eight datasets and eight methods ($B = 50$). Results are reported as mean (standard deviation). Best and second-best results are shown in **bold** and underlined, respectively. N/A indicates runs omitted due to prohibitive runtime.

| Method | Aliphatic Index | GRAVY Index | $\alpha$-helix | TF1 | TF2 | TF3 | AAV | Thermostability |
|---|---|---|---|---|---|---|---|---|
| BO | 551.45 (11.43) | 15.27 (1.44) | 1.91 (0.08) | 2.92 (0.13) | 2.88 (0.06) | 3.47 (0.04) | 2.35 (0.18) | 6.79 (0.02) |
| CMA-ES | 689.86 (19.16) | 20.82 (0.70) | 3.42 (0.21) | 4.12 (0.27) | 4.65 (0.28) | 4.00 (0.47) | 3.46 (0.14) | 3.74 (0.01) |
| AdaLead | 996.55 (110.77) | 26.90 (3.27) | 5.71 (0.26) | 4.67 (0.42) | 5.20 (0.30) | 4.13 (0.79) | 3.59 (0.14) | 6.46 (0.09) |
| Cbas | 447.97 (47.89) | 10.67 (0.38) | 2.22 (0.28) | 3.03 (0.02) | 2.73 (0.18) | 3.25 (0.10) | 2.27 (0.11) | 5.45 (1.01) |
| Dbas | 508.31 (25.60) | 9.92 (0.51) | 2.39 (0.57) | 3.19 (0.14) | 2.87 (0.15) | 3.17 (0.13) | 2.16 (0.27) | 5.38 (0.57) |
| GFN-AL | 217.18 (47.58) | 9.39 (0.39) | 1.44 (0.36) | 3.57 (0.15) | 4.77 (0.38) | 4.58 (0.25) | N/A | N/A |
| DynaPPO | 188.91 (11.39) | 15.72 (0.96) | 2.23 (0.38) | 3.30 (0.02) | 4.08 (0.02) | 3.90 (0.02) | N/A | N/A |
| IDEAS | **1639.84** (164.34) | **37.36** (2.45) | 5.83 (0.21) | **4.72** (0.37) | **5.82** (0.24) | 4.84 (0.60) | **4.37** (0.30) | **8.90** (0.21) |
| IDEAS-X | 1592.69 (190.97) | 35.10 (0.94) | **5.96** (0.24) | 4.68 (0.40) | 5.77 (0.49) | **5.31** (0.25) | 3.85 (0.21) | 8.40 (0.13) |

Despite the substantial progress in using explainable models to identify critical and non-salient motifs, existing work has largely focused on *post-hoc explainability* and biological insight. To the best of our knowledge, no prior method has systematically leveraged attribution scores from XAI models to *actively guide the biological sequence design process*. This is exactly the gap that motivates our proposed method, IDEAS.

## 5. Experimental Results

We evaluate IDEAS against **seven baseline** methods spanning evolutionary algorithms, reinforcement learning, and generative models across **eight continuous-property optimization tasks**. To assess performance under limited oracle budgets, a critical constraint in experimental settings, we compare methods at budget levels $B \in \{20, 50, 100\}$. Performance is measured using the AUC metric (Equation 10), which quantifies each method's ability to discover high-property sequences with minimal oracle queries. Beyond optimization performance, we analyze the diversity-performance and novelty-performance trade-offs through AUC-Diversity and AUC-Novelty plots, revealing each method's ability to balance exploitation with exploration of the sequence space.

**Datasets and Oracle functions.** We evaluate IDEAS on eight optimization tasks spanning different biological sequence types. The first two tasks optimize the **Aliphatic Index** and **GRAVY Index** of Anti-Cancer Peptides (ACPs), which are sequences of 20 amino acid types ($|\mathcal{V}| = 20$) with lengths ranging from 20 to 97 residues. These properties are particularly suitable for validation because their analytical expressions are known (see Appendix B), providing exact oracle evaluations and enabling qualitative analysis of the optimization process, as discussed in subsequent sections.

The third task optimizes the $\alpha$-**helix ratio** in protein primary sequences. The $\alpha$-helix ratio is defined as the fraction of sequence positions that form an $\alpha$-helical secondary

structure, normalized by the sequence length. We use the dataset of 3,655 protein sequences curated by Gupta & Zou (2019b) from the UniProt database (uni, 2017), with sequence lengths ranging from 20 to 50 residues and $|\mathcal{V}| = 20$. We employ the widely-used PSIPRED server (Buchan et al., 2013; Gupta & Zou, 2019b) as the oracle function.

The final three tasks involve optimizing the binding affinity of transcription factors (TFs) to DNA sequences of length 8, composed of the four canonical nucleotides (A, C, G, and T; $|\mathcal{V}| = 4$). Barrera et al. (2016b) compiled experimentally measured binding affinities for hundreds of TFs across all possible 8-mer DNA sequences, thereby providing exact oracle evaluations. We randomly select three TFs from this dataset, referred to as **TF1**, **TF2**, and **TF3** henceforth. Additional details are provided in Appendix B.

The other two tasks evaluate IDEAS on two challenging protein fitness landscapes from the FLIP benchmark (Dallago et al., 2021): **AAV fitness** and **Thermostability**. The AAV dataset consists of Adeno-Associated Virus (AAV) capsid protein sequences of length 734–750 residues ($|\mathcal{V}| = 20$), where fitness measures the viability of the capsid for gene therapy applications. The Thermostability dataset comprises protein sequences of varying lengths ($L \in [88, 5388]$, $|\mathcal{V}| = 20$), where the target property is the melting temperature ($T_m$), a direct measure of protein thermal stability. Both datasets represent complex fitness landscapes with long sequences. We employ ESM-2 (Lin et al., 2023) followed by pooling and a linear layer as the oracle model for both datasets, achieving a Pearson correlation of 0.89 on the AAV fitness dataset and a Spearman correlation of 0.70 on the Thermostability dataset.

**Baselines.** We compare IDEAS against a range of evolutionary, reinforcement learning, and generative baselines. For evolutionary methods, we include **AdaLead** (Sinai et al., 2020), which performs greedy hill climbing in sequence space, and **CMA-ES** (Hansen, 2006), which optimizes a continuous embedding of one-hot sequence encodings while

adaptively learning a full covariance matrix over sequence dimensions. As an RL-based baseline, we evaluate **DynaPPO** (Angermueller et al., 2019), which applies on-policy proximal policy optimization to sequence design. We also include **GFN-AL** (Jain et al., 2022), a GFlowNet-based approach (Bengio et al., 2021) that learns a stochastic generative policy and samples diverse high-property sequences via flow-matching objectives. In addition, we compare against **Bayesian Optimization (BO)** adapted to large biological sequence spaces following Sinai et al. (2020). Finally, we include two VAE-based generative models, **Dbas** (Brookes & Listgarten, 2018) and **Cbas** (Brookes et al., 2019), which use adaptive probabilistic sampling to progressively bias generation toward the objective function.

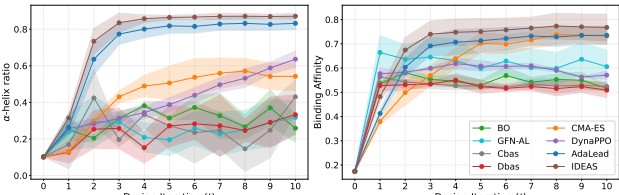

*Figure 3.* Evolution of the property $y$ as a function of design iteration ($t$) for a) the $\alpha$-helix ratio and b) the Binding affinity (TF1) datasets for $B = 100$.

**Evaluation Strategy.** For all datasets, we first curate an initial dataset $D_0 = \{x_i \mid y_i < y_{\text{cut}}\}$, where the cutoff values $y_{\text{cut}}$ for each dataset are provided in Table 5 in Appendix B. Starting from $\mathcal{D}_0$, we perform 10 design iterations ($t$) for each of three oracle function budgets, $B \in \{20, 50, 100\}$. For each method and budget, we conduct 10 independent trials and report results as the mean and standard deviation across these trials.

To isolate the effect of exploration in IDEAS, as discussed in Section 3, we also evaluate a restricted variant, denoted IDEAS-X, which performs only a single exploitative mutation with $\tau_1 = \tau_2 = 1$ and disables exploratory mutations.

### 5.1. IDEAS offers acceleration in sequence design

Starting from the initial dataset $\mathcal{D}_0$, we perform ten design iterations to optimize the target properties. Figures 3 and Appendix C.1 show the evolution of the mean property value of the generated sequences as a function of the design iteration $t$. Across all datasets and oracle budgets $B$, IDEAS consistently achieves faster improvements than competing baselines. We quantify this advantage using the area under the curve (AUC) of the corresponding trajectories; results are summarized in Table 1 for $B = 50$, in Table 8 for $B = \{20, 100\}$. **Overall, IDEAS outperforms all baselines by an average of 19%**, demonstrating its effectiveness for biological sequence optimization. Comparing IDEAS with its restricted variant IDEAS-X, we observe that their

mean AUC values differ by less than 2%, indicating that exploitative mutations account for the majority of the performance gains in IDEAS. The contribution of exploratory mutations is therefore not reflected in the AUC metric and is instead examined in our analysis of diversity and novelty.

To investigate the source of IDEAS's superior performance, we analyze the normalized frequency of all exploitative mutations for the GRAVY index, shown in Figure 4a. The figure reveals that the majority of mutations favor substitutions to amino acids I and L, with E→I and E→L among the most frequent transitions. This pattern is consistent with the analytical form of the GRAVY index presented in Appendix B, which assigns positive contributions to I and L and a negative contribution to E.

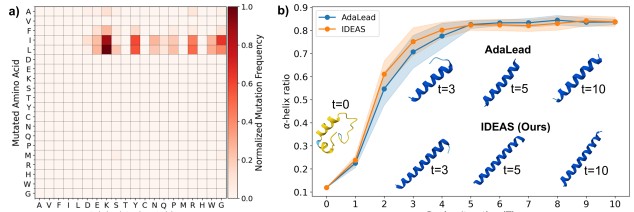

*Figure 4.* (a) Normalized frequency of all possible $20 \times 20$ mutations for the GRAVY index. (b) Structural evolution on the $\alpha$-helix ratio dataset comparing IDEAS with AdaLead (second-best).

Figure 4b shows the evolution of the $\alpha$-helix ratio as a function of $t$, along with the structures of the best sequences at selected iterations. We compare IDEAS with AdaLead, the second-best method. After $t = 5$, the best sequence produced by both the methods is largely helical. Moreover, **IDEAS achieves a best sequence with a 10% higher $\alpha$-helix ratio than AdaLead**, highlighting its ability to accelerate the sequence design process.

### 5.2. IDEAS performance lies on the Pareto curve

As noted by Jain et al. (2022), biological sequence design requires balancing AUC, diversity, and novelty. Accordingly, Figure 5 presents the AUC–Diversity trade-off for an oracle budget of $B = 50$, while results for $B \in 20, 100$ are shown in Figure 14 in the Appendix. The plots report the top five methods ranked by AUC in Table 1. Across budgets, IDEAS lies on the Pareto frontier, indicating a favorable balance between AUC and diversity. Although IDEAS and IDEAS-X achieve comparable AUC, **IDEAS attains 41% higher diversity**, attributable to its exploratory mutation strategy.

Similarly, Figure 15 in the Appendix presents the AUC–Novelty trade-off. In this plot, either IDEAS or IDEAS-X lies on the Pareto frontier, with less than a 2% difference in novelty between the two methods. Taken together, the AUC, Diversity, and Novelty analyses show that

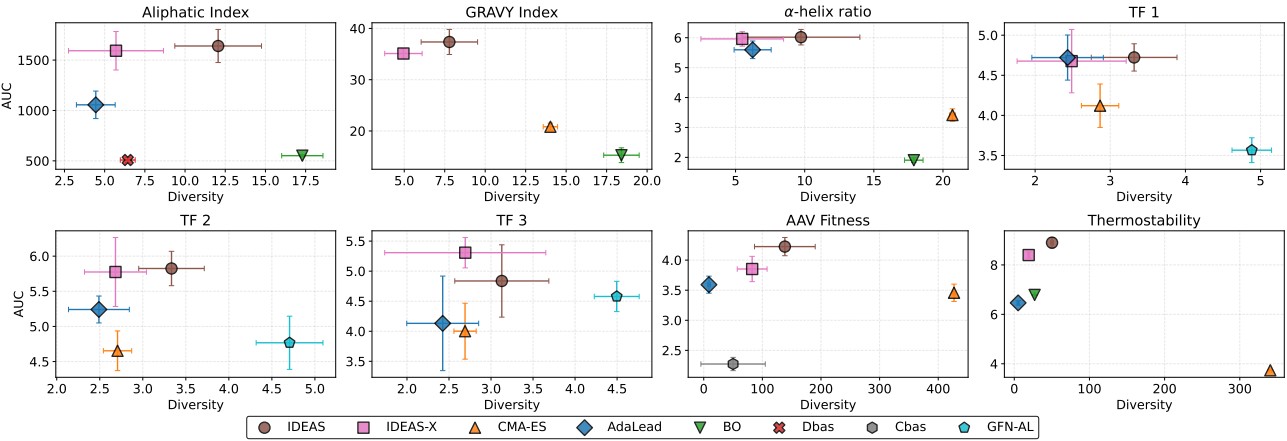

*Figure 5.* Trade-off between diversity and AUC across eight datasets with $B = 50$. IDEAS consistently lies on the Pareto curve.

IDEAS maintains a favorable balance across all three metrics, while achieving a 19% improvement in AUC over the baselines.

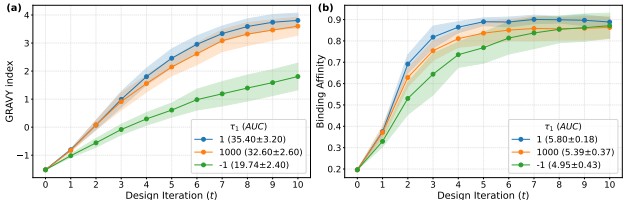

*Figure 6.* Impact of the position-selection temperature $\tau_1$ (Equation 6) on design performance for (a) GRAVY index and (b) Binding affinity (TF1).

### 5.3. Ablation Study

Having established the performance of IDEAS relative to baseline methods in terms of AUC, diversity, and novelty, we next assess the impact of individual components of IDEAS, including the temperature parameters ($\tau_1$ and $\tau_2$) and the number of mutations, on overall performance. For clarity in the ablation study, we disable the exploratory component of IDEAS to isolate the effects of these factors.

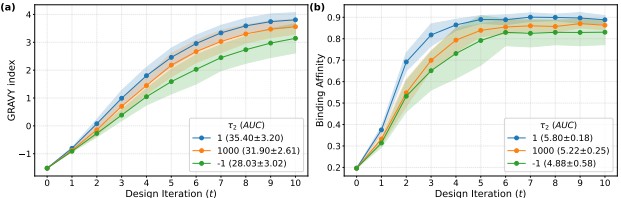

*Figure 7.* Impact of the replacement-motif selection temperature $\tau_2$ (Equation 8) on design performance for (a) GRAVY index and (b) Binding affinity (TF1).

#### 5.3.1. EFFECT OF TEMPERATURE ON DESIGN

From Equations 6–9, the selection of the mutation position ($j$) and the replacement motif (s) is governed by the temperature parameters $\tau_1$ and $\tau_2$, respectively. We therefore study their impact on overall design performance.

**Effect of $\tau_1$:** Figures 6a and 6b illustrate the effect of $\tau_1$ on optimizing the GRAVY index and binding affinity. Setting $\tau_1 = 1$ biases mutations toward positions with smaller $\psi_i$ (Equation 6), whereas $\tau_1 = -1$ favors positions with larger $\psi_i$. In contrast, $\tau_1 = 1000$ corresponds to uniform random position selection. For this study, we fix $\tau_2 = 1$, biasing IDEAS toward selecting replacement motifs with higher attribution scores. Across both tasks, $\tau_1 = 1$ yields average improvements of 8% and 48% over the $\tau_1 = 1000$ and $\tau_1 = -1$ settings, demonstrating that informed selection of mutation positions substantially influences sequence design performance. Diversity and novelty values for all cases remain within 1% of each other.

**Effect of $\tau_2$:** Analogous to the study of $\tau_1$, we evaluate $\tau_2 \in \{1, 1000, -1\}$ while fixing $\tau_1 = 1$. Setting $\tau_2 = 1$ biases IDEAS toward selecting replacement motifs with higher attribution scores, whereas $\tau_2 = -1$ favors motifs with lower attribution scores. Figures 7a and 7b show the effect of $\tau_2$ on the GRAVY index and binding affinity, respectively. Across both tasks, $\tau_2 = 1$ achieves average improvements of 11% and 22% over the $\tau_2 = 1000$ and $\tau_2 = -1$ settings, underscoring the importance of replacement motif selection during mutation.

For the GRAVY index, setting $\tau_1 = 1$ improves performance by 79% over $\tau_1 = -1$, while $\tau_2 = 1$ yields a 26% improvement over $\tau_2 = -1$. In contrast, for binding affinity, $\tau_1 = 1$ provides a 17% improvement, whereas $\tau_2 = 1$ provides a 19% improvement over their respective negative settings. These results indicate that mutation position selection (controlled by $\tau_1$) has a larger impact on GRAVY optimization, while replacement motif selection (controlled by $\tau_2$) plays a comparatively greater role in binding affinity optimization. Overall, neither component alone dominates sequence design performance; prioritizing one over the other can lead to decelerated design process.

**Varying $\tau_1$ and $\tau_2$ together:** Having examined the effects of $\tau_1$ and $\tau_2$ independently, we next study how design performance varies as the mutation strategy transitions from highly exploitative ($\tau_1, \tau_2 = 0.05$) to highly exploratory ($\tau_1, \tau_2 = 1000$). The AUC–Diversity trade-off in Figure 8a shows that AUC decreases as temperature increases and plateaus for $\tau_1, \tau_2 \geq 5$, while the maximum AUC is reached around $\tau_1, \tau_2 \approx 0.1$. In contrast, diversity increases with temperature, reaching its minimum near $\tau_1, \tau_2 \approx 0.1$ and saturating around $\tau_1, \tau_2 \approx 5$. These results indicate that increased exploration slows optimization while promoting diversity, highlighting the trade-off between design efficiency and diversity.

### 5.3.2. Effect of Number of Mutations of Design

Based on the description of IDEAS in Section 3, we perform a single exploitative mutation by default; for ablation studies, the exploratory component is disabled. In this experiment, we examine the effect of the number of exploitative mutations ($\mu$) on design performance while fixing $\tau_1 = \tau_2 = 1$.

For $\mu > 1$, we sample $\mu$ unique mutation positions from $x_i$ using Equation 9a without replacement, and select $\mu$ replacement motifs using Equation 9b with replacement. This choice reflects the fact that multiple positions can be mutated using the same motif. Figure 8b presents the resulting AUC-Diversity trade-off as a function of $\mu$. As $\mu$ increases, AUC improves while diversity decreases, indicating a trade-off between optimization performance and diversity. Mutating multiple positions with similar motifs reduces sequence variability, leading to lower diversity.

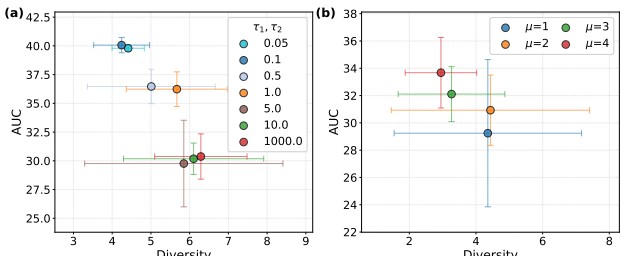

*Figure 8.* Impact of a) Temperature $\tau_1, \tau_2$, and b) the number of mutations ($\mu$) on design performance.

### 5.3.3. IDEAS can be integrated with different XAI models

To demonstrate that IDEAS is agnostic to the choice of XAI, we integrate five representative explainability methods—COLOR, DeepLift, DeepLiftShap, GradientSHAP, and Integrated Gradients—into the IDEAS framework and evaluate their performance on optimizing the GRAVY and Aliphatic indices. These properties are well-suited for this analysis because their oracle functions $f(x_i)$ are analytical expressions, enabling direct assessment of the faithfulness of the resulting attribution scores $\phi_i$. In this setting, the at-

tribution scores produced by all five explainability methods are equally faithful to the ground-truth contributions. As shown in Figure 9, all methods achieve nearly identical optimization performance, with average AUC values differing by less than 2%. These results indicate that IDEAS is robust to the choice of XAI model, provided the attribution method yields faithful explanations.

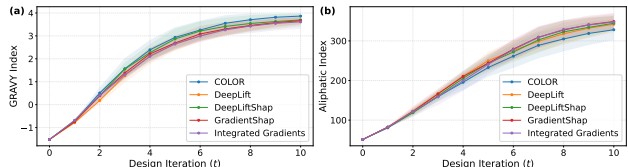

*Figure 9.* Effect of different XAI models with similar faithful attribution scores on optimizing (a) the GRAVY index and (b) the Aliphatic index.

### 5.3.4. Robustness to Attribution Noise

We evaluate the sensitivity of IDEAS to attribution quality $\phi$ by injecting noise into training labels of the GRAVY index dataset:

$$Y_{\text{noisy}} = Y + \mathcal{N}\left(0, (\sigma(Y) \cdot \eta)^2\right), \tag{11}$$

where $\eta$ controls the noise level and $\phi$ quality is measured via Pearson correlation (PCC) with ground truth attribution scores. The results, presented in Table 2, demonstrate two key robustness properties: (1) robust to attribution noise: a 66% drop in PCC ($\eta = 0 \to 10$) leads to only an 8% reduction in AUC; (2) robust to noisy training data: even at large noise ($\eta = 10$), AUC remains above the second-best baseline (AdaLead, AUC $= 26.9$). Performance degrades to AdaLead's level only when $\phi$ is effectively random ($\eta = 50$).

*Table 2.* Effect of label noise $\eta$ on attribution quality (PCC) and design performance (AUC) on the GRAVY index dataset.

| $\eta$ | PCC | AUC (mean $\pm$ std) |
|---|---|---|
| 0 | 0.99 | $37.10 \pm 1.97$ |
| 0.5 | 0.97 | $36.77 \pm 1.71$ |
| 1 | 0.91 | $37.04 \pm 2.00$ |
| 2 | 0.87 | $35.73 \pm 2.55$ |
| 5 | 0.76 | $36.42 \pm 2.68$ |
| 10 | 0.60 | $34.22 \pm 2.29$ |
| 50 | $-0.1$ | $26.44 \pm 4.79$ |

## 6. Conclusion

In this work, we introduced IDEAS, an interpretable evolutionary framework for biological sequence design that integrates explainable models (XAI) with evolutionary optimization to accelerate design under constrained oracle budgets. By leveraging attribution-informed mutations, IDEAS replaces random perturbations with interpretable mutations,

resulting in substantially improved sample efficiency over conventional evolutionary methods. Moreover, `IDEAS` achieves faster design progress than reinforcement learning– and generative model–based approaches by effectively exploiting attribution scores without requiring large training datasets. Across eight design tasks and multiple oracle budgets, `IDEAS` achieves an average 19% acceleration over seven competitive baselines while consistently operating on the Pareto frontier of AUC–Diversity and AUC–Novelty trade-offs.

**Limitations and Future Work.** While `IDEAS` demonstrates strong design acceleration, its diversity and novelty are lower than those of RL- and generative model–based approaches, reflecting the inherent locality of evolutionary mutations. In `IDEAS`, exploitative and exploratory mutations induce only controlled deviations from parent sequences, resulting in comparatively lower diversity and novelty. An important direction for future work is the integration of attribution-guided signals within RL or generative frameworks, combining the sample efficiency and interpretability of `IDEAS` with the global exploration capabilities of learned sequence generators.

## Software and Data

All source code to reproduce experimental results is available at https://github.com/pandeyakash23/IDEAS. We use public datasets and include implementation details in the Appendix. All baselines either adopt published hyperparameters or are tuned when unspecified.

## Impact Statement

This work addresses biological sequence design, a central problem in biotechnology and therapeutic development. We introduce `IDEAS`, which enables efficient sequence design under limited data and tightly constrained oracle budgets—a setting common in time-consuming and costly experimental and computational workflows. While this study focuses on biological sequences, the proposed explainability-driven evolutionary framework is broadly applicable to other materials design problems, such as polymer design.

## Acknowledgment

We gratefully acknowledge the support of the National Science Foundation's MRSEC program (DMR-2308691) at the Materials Research Center of Northwestern University.

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

# Appendix

## A. COLOR: XAI Model

COLOR, developed by Pandey et al. (2025), computes position-wise attribution scores using an **absolute** value operation. This approach assigns equal positive importance to positions that contribute positively or negatively to the property of interest. For sequence design, this is suboptimal, as it is crucial to distinguish between positively and negatively contributing positions and motifs. In the present study, we replace the absolute operation with a **ReLU** function, enabling IDEAS to focus exclusively on positively attributing positions and motifs.

All datasets considered in this work consist of continuous properties; therefore, we use mean squared error (MSE) as the loss function and train the COLOR model using the Adam optimizer. Model training is stopped based on the validation loss. The number of training parameters in the model for each dataset is given in Table 3. For all datasets except GRAVY and Aliphatic index, positional encodings are used to inject position information into the residue representations.

| Dataset | Number of parameters in COLOR model |
|---|---|
| Aliphatic index | 10,260 |
| GRAVY index | 10,260 |
| $\alpha$-helix ratio | 14,440 |
| TF1, TF2, TF3 | 9,981 |
| AAV Fitness | 43,005 |
| Thermostability | 43,851 |

*Table 3.* Number of training parameters in the COLOR model for different datasets.

## B. Dataset Details

### B.1. Aliphatic Index

The aliphatic index is a physicochemical property of a protein sequence that quantifies the relative volume occupied by aliphatic side chains. It is defined as the weighted sum of the mole fractions of aliphatic amino acids—alanine (A), valine (V), isoleucine (I), and leucine (L)—in the sequence. Formally, for a sequence $x$ of length $L$, the aliphatic index is given by

$$\text{AI}(x) = \chi_A + a\,\chi_V + b\,(\chi_I + \chi_L), \tag{12}$$

where $\chi$ denotes the mole fraction of the corresponding amino acid in $x$, and $a = 2.9$, $b = 3.9$ are empirically determined coefficients. Due to its analytical formulation, the aliphatic index admits an exact oracle function, making it well-suited for evaluating the faithfulness of attribution scores and the effectiveness of sequence design methods.

### B.2. GRAVY index

**GRAVY Index.**  The grand average of hydropathy (GRAVY) index measures the overall hydrophobicity of a protein sequence and is computed as the average of hydropathy values of its constituent amino acids. For a sequence $x$ of length $L$, the GRAVY index is given by $\text{GRAVY}(x) = \frac{1}{L}\sum_{i=1}^{L} h(x_i)$, where $h(\cdot)$ (Table 4) denotes the hydropathy score of an amino acid. Similar to the aliphatic index, the GRAVY index admits an analytical oracle, enabling exact evaluation and faithful attribution analysis.

## C. Additional Results

### C.1. Evolution of property as a function of design iterations

Figure 11 and Figure 12 compare the evolution of the target property $y$ as a function of design iterations across eight methods on eight different datasets for oracle budget $B =$ 100, 50, and 20, respectively.

### C.2. Effect of motif size

Figure 13 illustrates the effect of the motif size $m$ in Equation 4 on design performance. Increasing the motif size from $m = 1$ to $m = 2$ leads to substantial gains, with AUC and Diversity improving by 23% and 36%, respectively. Beyond this

*Table 4.* Hydropathy values for amino acids.

| Amino Acid | Hydropathy Value |
|---|---|
| Alanine (A) | 1.8 |
| Arginine (R) | -4.5 |
| Asparagine (N) | -3.5 |
| Aspartic acid (D) | -3.5 |
| Cysteine (C) | 2.5 |
| Glutamine (Q) | -3.5 |
| Glutamic acid (E) | -3.5 |
| Glycine (G) | -0.4 |
| Histidine (H) | -3.2 |
| Isoleucine (I) | 4.5 |
| Leucine (L) | 3.8 |
| Lysine (K) | -3.9 |
| Methionine (M) | 1.9 |
| Phenylalanine (F) | 2.8 |
| Proline (P) | -1.6 |
| Serine (S) | -0.8 |
| Threonine (T) | -0.7 |
| Tryptophan (W) | -0.9 |
| Tyrosine (Y) | -1.3 |
| Valine (V) | 4.2 |

*Table 5.* Dataset parameters showing cutoff values ($y_{cut}$) and initial sample sizes ($N_0$).

| Dataset | Cutoff ($y_{cut}$) | $N_0$ |
|---|---|---|
| Aliphatic index | 20 | 50 |
| GRAVY index | 0.14 | 100 |
| $\alpha$-helix ratio | 0.15 | 200 |
| TF1, TF2, TF3 | 0.2 | 250 |
| AAV Fitness | 0.2 | 500 |
| Thermostability | 0.2 | 700 |

point, both metrics exhibit diminishing returns and largely plateau as $m$ increases further. Novelty also shows a modest increase with larger motifs, though the improvement is comparatively limited. Overall, this analysis indicates that while an initial increase in motif size can significantly enhance design performance, larger motifs provide marginal additional benefits.

### C.3. Wall-clock time comparison

Wall-clock time is measured on the GRAVY index dataset, where the oracle is given by an analytical expression. As shown in Table 7, IDEAS occupies a middle ground between lightweight evolutionary methods (AdaLead, CMA-ES) and deep generative models (Cbas, Dbas), while being $16\times$ and $50\times$ faster than GFN-AL and DynaPPO, respectively. All models are run on an AMD EPYC 7413 24-Core Processor for this study.

### C.4. Choice of Motif Size

In case of the AAV dataset (Dallago et al., 2021), in IDEAS, the motif size $m$ is selected based on the COLOR (Pandey et al., 2025) model, which partitions a sequence into overlapping motifs of length $m$ and learns a global representation by aggregating their latent embeddings. In practice, $m$ is chosen to maximize COLOR's predictive performance (e.g., PCC, $R^2$, MSE) on a validation set, and this optimal $m$ is carried forward into IDEAS. Table 6 reports the effect of $m$ on both COLOR's predictive performance (Pearson correlation coefficient; PCC) and IDEAS's AUC on the AAV dataset, where both metrics peak at $m=10$, empirically validating our criterion for selecting $m$.

*Table 6.* Effect of motif size $m$ on COLOR's predictive performance (PCC) and `IDEAS`'s AUC on the AAV dataset.

| Motif Size ($m$) | PCC | AUC |
|---|---|---|
| 1 | 0.86 | $3.90 \pm 0.21$ |
| 3 | 0.87 | $3.92 \pm 0.34$ |
| 6 | 0.89 | $3.97 \pm 0.15$ |
| 10 | **0.92** | **$4.37 \pm 0.30$** |
| 15 | 0.86 | $3.70 \pm 0.40$ |

| Method | BO | CMA-ES | AdaLead | Cbas | Dbas | GFN-AL | DynaPPO | IDEAS |
|---|---|---|---|---|---|---|---|---|
| **Runtime (s)** | 6.2 | 0.033 | 0.003 | 2.8 | 3.1 | 21 | 65 | 1.3 |

*Table 7.* Runtime comparison (in seconds) across different methods.

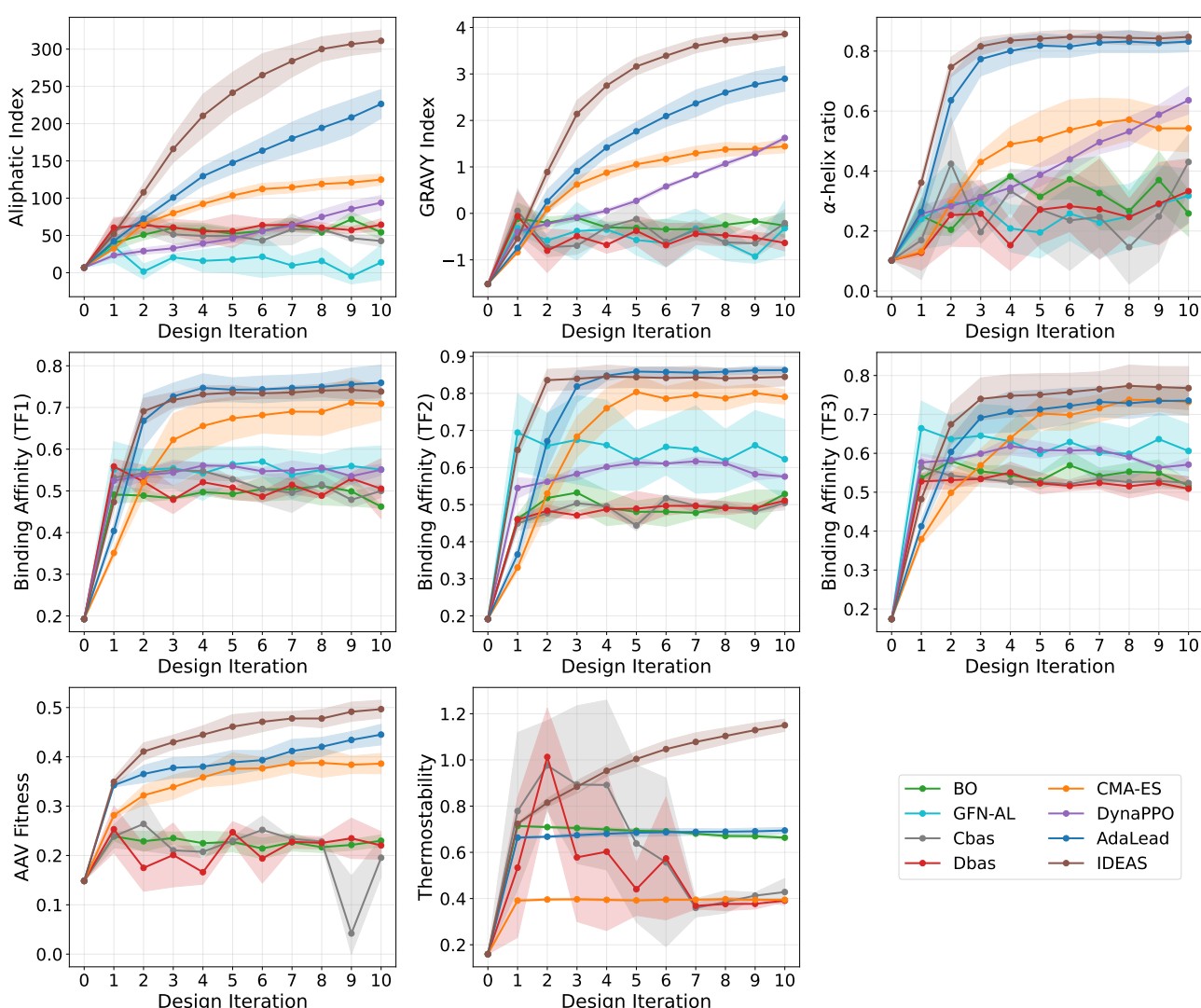

*Figure 10.* Evolution of property $y$ shown as a function of design iteration ($t$) for all the datasets and $B = 100$.

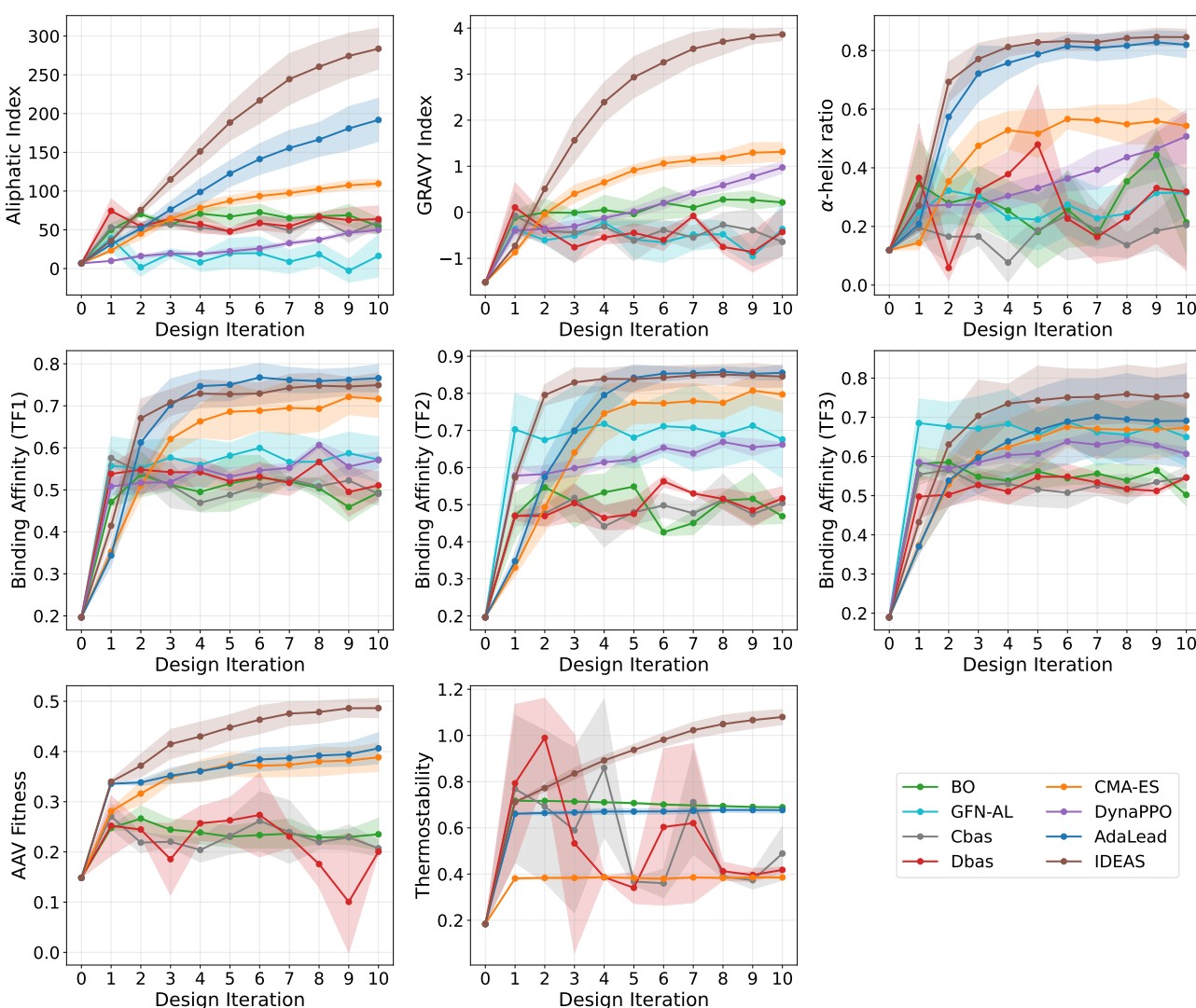

*Figure 11.* Evolution of property $y$ shown as a function of design iteration ($t$) for all the datasets and $B = 50$.

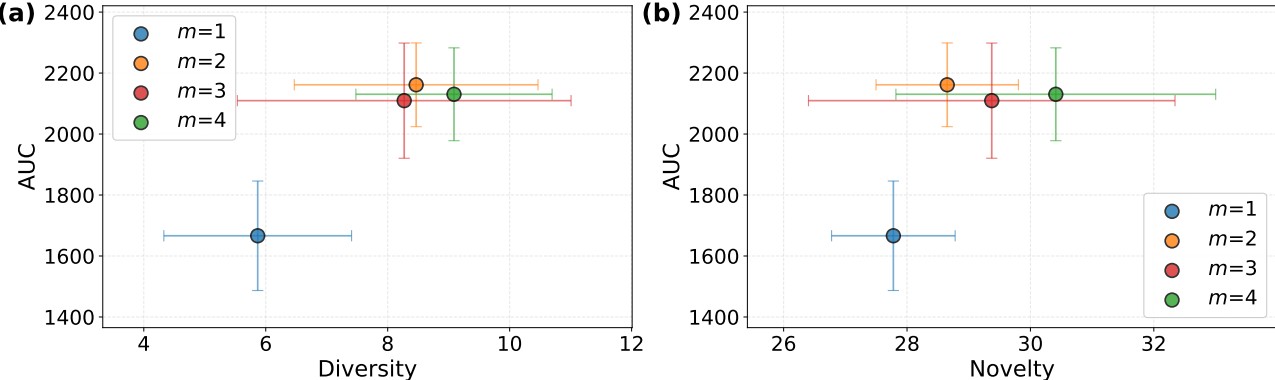

*Figure 12.* Evolution of property $y$ shown as a function of design iteration ($t$) for all the datasets and $B = 20$.

*Figure 13.* Effect of motif size ($m$) shown on Aliphatic index dataset using a) AUC-Diversity and b)AUC-Novelty plot.

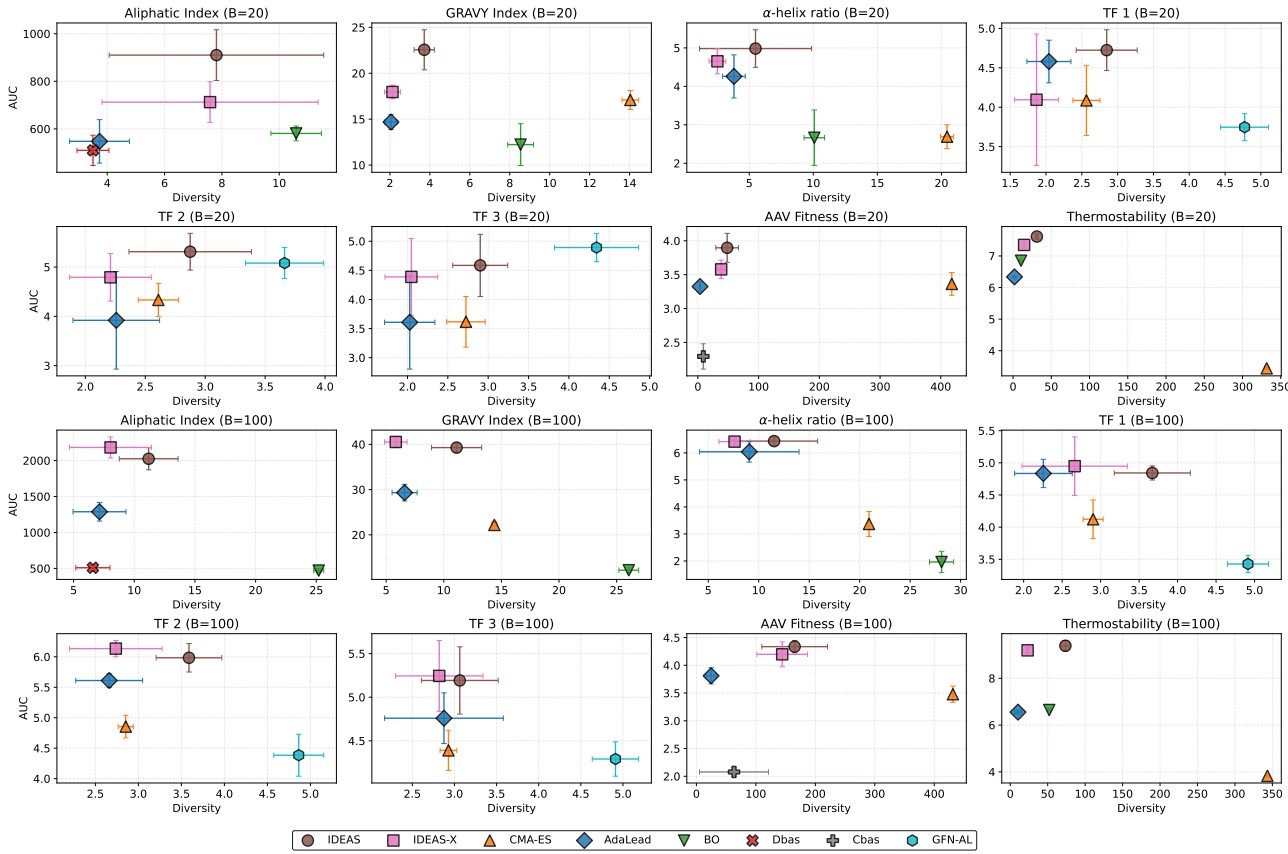

*Figure 14.* Trade-off between Diversity and performance (AUC) across eight different dataset for $B = \{20, 100\}$.

*Table 8.* Area under the curve (AUC) comparison across eight datasets and eight methods for $B \in \{20, 100\}$. Results are reported as Mean (standard deviation). Best results are shown in **bold** and second best are underlined. N/A indicates runs omitted due to prohibitive runtime.

| Method | Aliphatic Index 20 | Aliphatic Index 100 | GRAVY Index 20 | GRAVY Index 100 | α-helix 20 | α-helix 100 | TF1 20 | TF1 100 | TF2 20 | TF2 100 | TF3 20 | TF3 100 | AAV 20 | AAV 100 | Thermostability 20 | Thermostability 100 |
|---|---|---|---|---|---|---|---|---|---|---|---|---|---|---|---|---|
| BO | 580.90 (30.97) | 469.95 (28.55) | 12.23 (2.28) | 12.18 (0.54) | 2.66 (0.72) | 1.06 (0.39) | 2.93 (0.19) | 2.07 (0.02) | 3.20 (0.13) | 2.87 (0.03) | 3.24 (0.13) | 3.56 (0.08) | 2.77 (0.30) | 2.22 (0.14) | 6.85 (0.02) | 6.64 (0.05) |
| CMA-ES | 450.62 (67.90) | 839.00 (39.78) | 17.09 (1.03) | 22.19 (0.75) | 2.69 (0.31) | 3.37 (0.46) | 4.09 (0.44) | 4.12 (0.30) | 4.33 (0.34) | 4.85 (0.19) | 3.62 (0.44) | 4.39 (0.23) | 3.36 (0.17) | 3.48 (0.15) | 3.44 (0.02) | 3.83 (0.01) |
| AdaLead | 556.11 (97.02) | 1341.10 (146.38) | 13.49 (1.02) | 31.00 (2.40) | **4.79** (0.43) | 5.15 (0.20) | 4.30 (0.46) | 4.40 (0.29) | 4.40 (0.49) | 5.43 (0.11) | 3.61 (0.80) | 4.70 (0.29) | 3.33 (0.07) | 3.81 (0.15) | 6.34 (0.02) | 6.55 (0.09) |
| Cbas | 456.59 (27.65) | 437.65 (60.27) | 6.69 (1.62) | 10.30 (0.57) | 2.22 (0.04) | 1.91 (0.36) | 3.24 (0.22) | 3.11 (0.11) | 2.81 (0.88) | 2.79 (0.07) | 3.29 (0.13) | 3.41 (0.02) | 2.29 (0.19) | 2.08 (0.08) | 5.30 (0.39) | 6.19 (0.47) |
| Dbas | 509.73 (63.71) | 509.40 (40.09) | 6.89 (0.38) | 9.60 (0.84) | 1.81 (0.25) | 1.90 (0.18) | 3.09 (0.11) | 3.03 (0.09) | 2.76 (0.11) | 2.80 (0.09) | 3.28 (0.18) | 3.35 (0.10) | 1.60 (0.30) | 2.11 (0.06) | 5.40 (0.59) | 5.14 (0.16) |
| GFN-AL | 198.07 (57.15) | 207.21 (75.68) | 9.64 (0.77) | 9.56 (0.64) | 1.40 (0.33) | 1.48 (0.36) | 3.75 (0.17) | 3.43 (0.13) | 5.08 (0.32) | 4.38 (0.34) | **4.89** (0.24) | 4.29 (0.20) | N/A | | N/A | |
| DynaPPO | 97.78 (22.30) | 432.46 (58.98) | 10.58 (0.26) | 18.64 (0.26) | 1.88 (0.33) | 2.99 (0.29) | 3.34 (0.07) | 3.36 (0.08) | 3.98 (0.14) | 3.80 (0.02) | 3.99 (0.08) | 3.99 (0.05) | N/A | | N/A | |
| IDEAS | **910.14** (106.55) | 2024.83 (153.63) | **22.56** (2.18) | 39.28 (1.05) | 4.75 (0.46) | **6.55** (0.13) | **4.72** (0.33) | 4.85 (0.22) | **5.31** (0.37) | 5.98 (0.23) | 4.59 (0.53) | 5.19 (0.39) | **3.90** (0.21) | **4.78** (0.20) | **7.62** (0.10) | **9.39** (0.23) |
| IDEAS-X | 712.44 (85.62) | **2182.40** (146.41) | 17.96 (0.74) | **40.53** (1.12) | 4.65 (0.33) | 6.42 (0.13) | 4.60 (0.83) | **4.95** (0.45) | 4.79 (0.48) | **6.13** (0.13) | 4.39 (0.66) | **5.24** (0.41) | 3.58 (0.14) | 4.20 (0.22) | 7.35 (0.10) | 9.20 (0.14) |

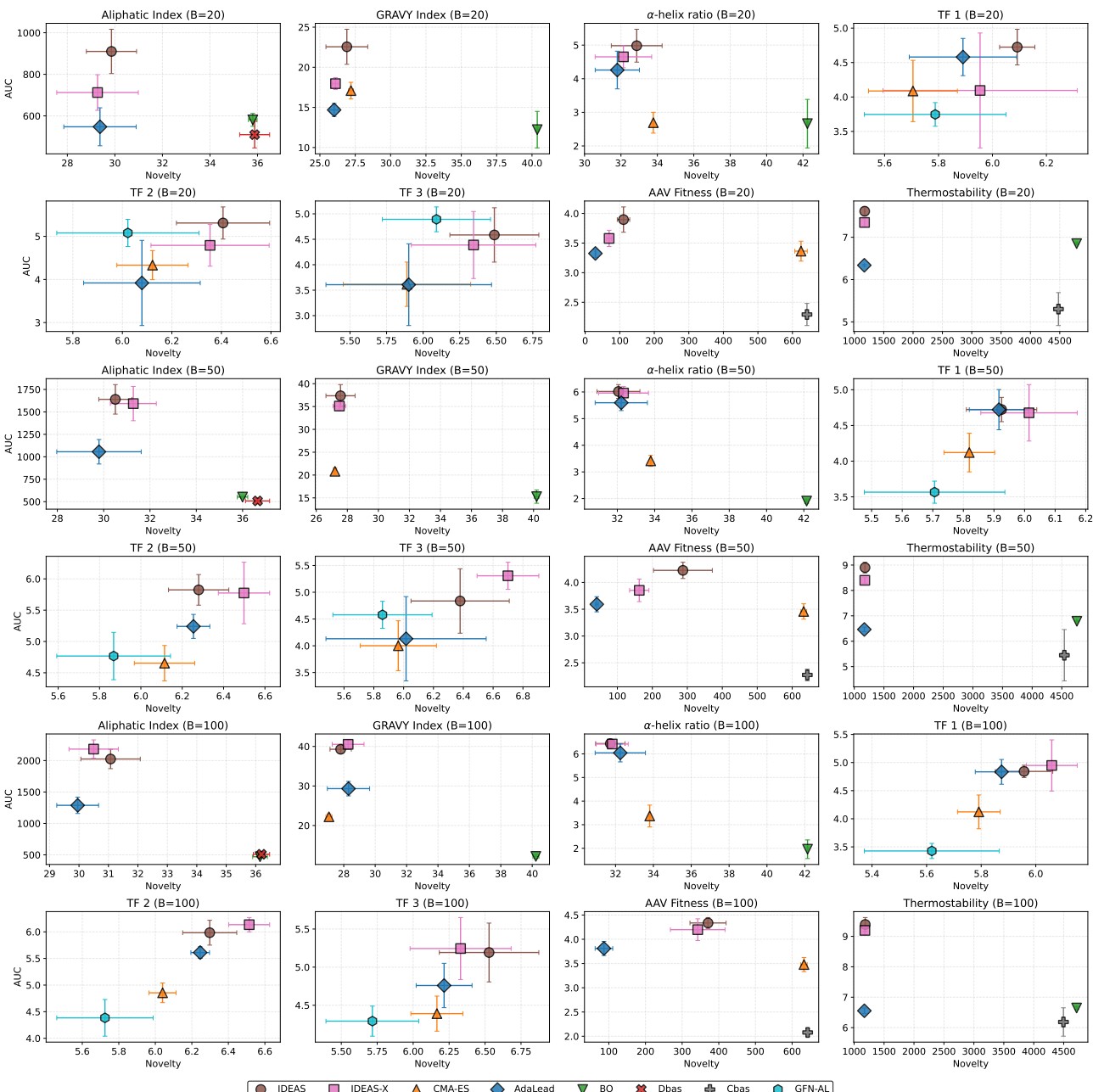

*Figure 15.* Trade-off between Novelty and performance (AUC) across eight different dataset for $B = \{20, 50, 100\}$.

