# OpenReview forum: "Interpretability Driven Evolutionary Approach for the Design of Biological Sequences"
_ICML.cc/2026/Conference — ICML 2026 regular_

### Official Review · Reviewer_Xuze · 2026-02-19

**Soundness:** 2
**Presentation:** 3
**Significance:** 2
**Originality:** 4
**Overall Recommendation:** 4
**Confidence:** 5

**Summary:**

This paper presents IDEAS, an interpretability-driven approach to biological sequence design. IDEAS learns an explainable model, identifying critical motifs that need to be conserved while mutating non-critical mutations. Across design tasks spanning biochemical properties and transcription factors, IDEAS achieves a 19% acceleration in design.

**Compliance With Llm Reviewing Policy:**

Affirmed.

**Final Justification:**

As I mentioned in my review, I appreciate the method, but was skeptical at their experimental setup. I believe my concerns have been thoroughly addressed, and I see no reason why I should not raise my score. I raise my score to a 4 in line with this.

**Key Questions For Authors:**

1. What was the paper's rationale for including simplistic and/or short biological sequence length landscapes?
2. What is the paper's rationale for using COLOR as the primary XAI framework? As I understand it, COLOR enables efficient computation of attribution values, but DeepSHAP/DeepLIFT are also fairly computationally efficient. Would the authors recommend using COLOR as the XAI model of choice for practical applications?
3. What does the interpretability aspect of IDEAS buy one in biological sequence design? Is it purely for accelerating sequence design, or are the mutations the XAI model suggests logical and could complement rational protein design? I'm particularly interested if you could elaborate specifically in the context of transcription factors.
4. Have the authors considered the effects of epistasis in biological sequence design? Attribution scores are fundamentally unable to capture such effects, while interaction scores are able to. Are there practical ramifications of this, or are its effects largely negligible as more design iterations happen?

**Limitations:**

The limitations section in the paper is thoughtfully written and appropriate, except for some points that I have raised above.

**Strengths And Weaknesses:**

Strengths
- The paper is clear and well-motivated.
- Adding an explainable (XAI) model for biological sequence design is an interesting and creative approach.
- The submission appears to be technically sound.
- The authors test a wide variety of baselines spanning representative methods from protein design literature.
- Ablation studies address appropriate questions such as relevant hyperparameters and choice of XAI model.
- IDEAS speeds up design with high sample efficiency, having the promise to address one of the primary bottlenecks in the field.

Weaknesses
- The main weakness of the paper is the experimental setup. While the authors test across six datasets, a significant portion of the datasets are too simplistic and not representative of actual design scenarios. For instance, optimizing for the Aliphatic index is a trivial task and can easily be solved by substituting out every amino acid for aliphatic residues, despite not being biologically feasible at all. The GRAVY index similarly can be solved for by substituting out every amino acid with hydrophobic residues.
- The design space tested is relatively small. For instance, the maximum length tested is around 100 amino acids, with most of these datasets being too simplistic, as previously mentioned. While the transcription factor experiments are particularly interesting, they are unfortunately capped to 8 nucleotides. As written, I have serious concerns whether IDEAS is able to extrapolate to more complex fitness landscapes, especially as the sequence length increases and epistatic (nonlinear) interactions compound. It would have been nice for the authors to add design experiments on proteins with significantly longer lengths, where function can be scored using an external in-silico evaluation model (with the caveat that you'd have to cap the number of mutations to 5-10 to maintain confident predictions). Such landscapes are significantly more difficult to optimize for, making it a more fair experiment to conduct.
- Implementation of baselines is lacking in the Appendix. The baselines as-is do not appear reproducible given the text.
- Minor issues with the text: T in Equation (10) appears to be undefined, making the AUC metric a little difficult to interpret. In Appendix B.1, "where · denotes the mole fraction" also appears to be a typo.

Overall, I believe that IDEAS is a well-motivated and creative method, but given the above weaknesses in its current state, I cannot recommend the paper for acceptance. Should the authors address my weaknesses and below questions adequately and convincingly, I am happy to consider raising my score.

---

> ### Author Rebuttal · Authors · 2026-03-30
>
> Thank you for the thoughtful review. We appreciate your recognition of IDEAS' clarity, technical soundness, and creativity. Below, we address all questions and comments; new experimental results will be incorporated into the revised paper.
>
>
> ## Weaknesses and Questions
>
> > (W1,W2, Q1: Datasets are simple):
>
> The datasets in the paper were selected for having exact oracle models. Based on reviewer feedback, we additionally evaluate on two FLIP [1] datasets: AAV fitness ($L\in[734,750]$) and Thermostability ($L\in[88,5388]$), using ESM-2 [2] based oracles (Pearson corr.: 0.89, Spearman corr.: 0.70). We evaluate AUC across oracle budgets $B\in${20,50,100}; results are shown in the table below, with $m=10$ used for both datasets. **IDEAS outperforms the next best baseline by an average of 21%.** DynaPPO and GFN-AL are excluded due to the long runtime (see W3/Q2, Reviewer TQjm).
>
> |Method|AAV(B=20)|AAV(B=50)|AAV(B=100)|Thermo(B=20)|Thermo(B=50)|Thermo(B=100)|
> |-|-|-|-|-|-|-|
> |BO|2.77±0.30|2.35±0.18|2.22±0.14|6.85±0.02|6.79±0.02|6.64±0.05|
> |CMA-ES|3.36±0.17|3.46±0.14|3.48±0.15|3.44±0.02|3.74±0.01|3.83±0.01|
> |AdaLead|3.33±0.07|3.59±0.14|3.81±0.15|6.34±0.02|6.46±0.09|6.55±0.09|
> |CBas|2.29±0.19|2.27±0.11|2.08±0.08|5.30±0.39|5.45±1.01|6.19±0.47|
> |DbAs|1.60±0.30|2.16±0.27|2.11±0.06|5.40±0.59|5.38±0.57|5.14±0.16|
> |IDEAS|**3.90±0.21**|**4.37±0.30**|**4.78±0.20**|**7.62±0.10**|**8.90±0.21**|**9.39±0.23**|
> |IDEAS-X|3.58±0.14|3.85±0.21|4.20±0.22|7.35±0.10|8.40±0.13|9.20±0.14|
>
> The above results confirm that IDEAS generalizes to complex fitness landscapes and long biological sequences. While wet-lab validation is beyond the scope of this work, it is an active direction we are currently pursuing through collaborative efforts.
>
> > (Q2: Rationale for using COLOR as XAI method)
>
> While IDEAS is compatible with any XAI method (Section 5.3.3), we use COLOR as the primary XAI backbone because its architecture provides a principled criterion for selecting motif size $m$: COLOR partitions sequences into motifs of length $m$ and learns a global representation by aggregating their latent embeddings, making $m$ an explicit choice optimized via predictive performance. As shown in our response to Q2 (Reviewer HbLk), both COLOR's predictive performance and IDEAS's AUC peak at $m{=}10$ in case of AAV dataset.
>
> In contrast, post-hoc methods (DeepSHAP, DeepLIFT, GradCAM, LIME) require a two-stage setup: surrogate model selection followed by attribution method selection, with no principled criterion for motif size.
>
> > (Q3: What does interpretability mean in case of IDEAS)
>
> IDEAS accelerates sequence design while suggesting biologically logical mutations. We illustrate this through two datasets.
>
> **$\alpha$-helix:** Via Equations 4--9, IDEAS identifies Ala, Glu, and Leu as top $\alpha$-helix formers and Gly, Pro, and Asn as top breakers, consistent with [3], suggesting mutations such as Gly$\to$Ala and Pro$\to$Glu.
>
> **Transcription Factor BCL6:** IDEAS identifies AT-rich motifs (AAT, TAA, TTA) as top contributors to binding affinity and CG-rich motifs (CCC, GGG, GCC) as negative contributors, consistent with [4]: AT-rich regions narrow the minor groove, enhancing affinity, whereas CG-rich regions widen it.
>
> Taken together, these results demonstrate that IDEAS not only accelerates sequence design but also perform bioloigically logical mutations.
>
> > (Q4: Effect of epistasis)
>
> Though attribution scores do not capture pairwise interactions, the practical impact of epistasis is largely negligible over iterative design. Table 1 shows IDEAS (1 exploitative + 1 exploratory mutation)  and IDEAS-X (1 exploitative mutation) AUC are within 2% of each other. Section 5.3.2 further shows that increasing number of exploitative mutations ($\mu$) from 1 to 4 improves AUC by 15%. Across all configurations, AUC does not decrease, empirically confirming that epistasis does not adversely affect IDEAS in practice. We also note a typo in Figure 8 caption: $n_m$ should be replaced with $\mu$.
>
> > (W3: Baseline implementation details)
>
> Full implementation details will be added to the paper.
>
> **DynaPPO:** Score threshold $\tau{=}0.5$; penalty score factor $\lambda{=}0.1$; exploration penalty radius $\epsilon{=}2$ (TF datasets) and $\epsilon{=}20$ (all other datasets); policy network learning rate $\gamma{=}0.0001$.
>
> **DbAs/CbAs:** Relaxation threshold $Q{=}0.9$; generative network: VAE.
>
> **GFN-AL:** Number of allowed mutations $K{=}4$ (TF datasets) and $K{=}40$ (all other datasets); acquisition function: Upper Confidence Bound (UCB); generations between proposals $G{=}5$ (TF datasets) and $G{=}50$ (all other datasets).
>
> > (W4: Typos)
>
> Thank you for pointing out. In Equation 10, $T$ denotes the total number of design iterations ($T{=}10$). In Appendix B.1, "." should be $\chi$. Both will be corrected in the revised paper.
>
>
> [1] Dallago et al., NeurIPS, 2021.
>
> [2] Lin et al, bioRxiv, 2022.
>
> [3] Pace et al, Biophysical Journal, 1998.
>
> [4] Rohs et al., Nature Articles, 2009.

---

> > ### Author Rebuttal · Reviewer_Xuze · 2026-04-01
> >
> > Thank you for the comments and clarifications. I appreciate the addition of the FLIP benchmarks, which alleviate my concerns. I am increasing my score according to a 4. I wish the authors the best of luck!

---

### Official Review · Reviewer_HbLk · 2026-03-12

**Soundness:** 3
**Presentation:** 3
**Significance:** 3
**Originality:** 3
**Overall Recommendation:** 4
**Confidence:** 4

**Summary:**

This paper proposed IDEAS framework, which addressed the inefficiency of random mutations in biological sequence design by integrating explainable AI (XAI)-generated attribution scores into evolutionary algorithms.
This approach utilizes XAI models to identify key motifs within high-fitness sequences, thereby directing mutations toward non-critical regions. Under constrained experimental budgets, it achieves an average 19% acceleration in design while maintaining a Pareto-optimal balance between performance, diversity, and novelty.

**Compliance With Llm Reviewing Policy:**

Affirmed.

**Final Justification:**

Most of my concerns are solved.

**Key Questions For Authors:**

- The paper states that IDEAS' effectiveness depends on the faithful attribution scores generated by XAI models. However, in practical black-box tasks, there are no analytical solutions like the GRAVY index for comparison. Can the authors provide a metric to evaluate attribution quality in tasks without analytical solutions?

- Authors achieved 19% acceleration at $m=1$ (monomer level) and noted limited gains from increasing $m$. However, biological sequence function often depends on specific local structural fragments. Is $m=1$ still sufficient to capture cooperative effects between amino acids in longer or more complex sequence tasks?

- How sensitive is IDEAS to the quality of the initial population?

**Limitations:**

please refer to the weakness

**Strengths And Weaknesses:**

## Strength
- High Efficiency: Under constrained experimental budgets, IDEAS achieves an average 19% design acceleration over existing benchmark methods by reducing blind search through informed mutation
- Balance between novelty and performance: The method achieves balance among AUC (performance), diversity, and novelty.
- Generality: IDEAS operates as a plug-and-play framework independent of specific interpretable models. Experiments demonstrate nearly identical performance to traditional XAI methods.
- Low computational overhead: Compared to generative models requiring extensive pre-training, IDEAS leverages evolutionary strategies that do not need massive initial datasets.

## Weaknesses
- As a variant of evolutionary algorithms, IDEAS' mutations primarily introduce controlled deviations near parent sequences. This results in slightly inferior global search capabilities and absolute diversity compared to reinforcement learning or flow-based generative models.
- Dependence on attribution quality: IDEAS' effectiveness heavily relies on the XAI model's ability to generate faithful attribution scores.
If the underlying prediction model produces erroneous attributions due to data noise or overfitting, it may guide the algorithm toward incorrect optimization directions.

---

> ### Author Rebuttal · Authors · 2026-03-30
>
> Thank you for your thoughtful review. We appreciate your recognition of IDEAS' efficiency gains, low computational cost compared to generative models, its balance between performance and diversity, and its plug-and-play generality across interpretable models. Below, we address your questions and comments. All new experimental results discussed below will be incorporated into the revised paper.
>
> ## Weaknesses (W) and Questions (Q)
> >(W1: IDEAS produces lesser diverse sequences than RL or flow-based methods):
>
> Evolutionary methods typically yield lower diversity than RL or flow-based approaches due to local mutation-based candidate generation. However, IDEAS compensates with superior sample efficiency: Table 1 shows IDEAS is $1.7\times$ and $2\times$ more sample-efficient than DynaPPO (RL) and GFN-AL (flow-based), respectively, under oracle budgets $B \in$ {20, 50, 100}. Moreover, Figures 5, 14, and 15 show that IDEAS maintains a position on the Pareto frontier in both AUC-Diversity and AUC-Novelty plots, demonstrating a superior balance across sample efficiency, diversity, and novelty under realistic budget constraints.
>
> >(W2: Dependence on attribution quality,$\phi$):
>
> We evaluate the sensitivity of IDEAS to the quality of $\phi$ by injecting noise into training labels:
> $Y_{\text{noisy}} = Y + \mathcal{N}(0, (\sigma(Y)\eta)^2),$
> where $\sigma(Y)$ is the standard deviation and $\eta$ controls the noise level. Experiments are conducted on the GRAVY index, where ground-truth $\phi$ is available, and attribution quality is measured via Pearson correlation (PCC) with the ground truth.
>
> Table below, shows the effect of $\eta$ on PCC and design performance (AUC). Notably, a 66\% drop in PCC ($\eta=0 \rightarrow 10$) leads to only an 8\% reduction in AUC, which remains higher than the second-best baseline (AdaLead, AUC = 26.9), demonstrating the robustness of IDEAS to noisy $\phi$. When $\phi$ becomes effectively random ($\eta=50$), performance drops to the level of AdaLead.
>
> | $\eta$ | PCC  | AUC (mean ± std) |
> |-------|------|------------------|
> | 0     | 0.99 | 37.10 ± 1.97     |
> | 0.5   | 0.97 | 36.77 ± 1.71     |
> | 1     | 0.91 | 37.04 ± 2.00     |
> | 2     | 0.87 | 35.73 ± 2.55     |
> | 5     | 0.76 | 36.42 ± 2.68     |
> | 10    | 0.60 | 34.22 ± 2.29     |
> | 50    | -0.1   | 26.44 ± 4.79     |
>
>
> >(Q1: Metric to evaluate attribution scores ($\phi$)):
>
> We thank the reviewer for the question. When ground-truth $\phi$ is unavailable, we use the metric from Pandey et al. [1], which evaluates attribution quality via progressive unmasking.
>
> Positions in $x \in \mathbb{R}^L$ are unmasked in descending order of $\phi_i$, and the loss $l(u)$ is measured as a function of the unmasked percentage $u$. The score is:
> $I = \int_{0}^{100} l(u)\,du,$
> where lower values indicate better attribution. This enables selecting the most effective $\phi$ within IDEAS.
>
>
> >(Q2: Is motif size m=1 enough for longer sequence lengths):
>
> We refer the reviewer to our response to W1/Q1 (Reviewer Xuze), where we evaluate IDEAS on two additional FLIP [2] datasets with longer sequences: AAV ($L\in[734\text{-}750]$) and Thermostability ($L\in[88\text{-}5388]$). IDEAS achieves 21% higher AUC than the best baseline at $m{=}10$, while reducing to $m{=}1$ drops AUC by 12%, confirming that larger motifs are necessary for complex fitness landscapes.
>
> In IDEAS, we select the motif size ($m$) based on the COLOR [1] model, which partitions a sequence into motifs of length $m$ and learns a global representation by aggregating their latent embeddings. In practice, $m$ is chosen to maximize COLOR's predictive performance (e.g., PCC, $R^2$, MSE) on a validation set, and this optimal $m$ is carried forward into IDEAS. The table below shows the effect of $m$ on both predictive performance (Pearson correlation; PCC) and IDEAS's AUC, where both metrics peak at $m{=}10$, empirically validating our criterion for selecting $m$.
>
> | Motif Size (m) | Predictive Performance of COLOR (PCC) | AUC |
> |----------------|----|-----|
> | 1  | 0.86 |3.90 ± 0.21 |
> | 3  | 0.87|3.92 ± 0.34 |
> | 6  | 0.89 |3.97 ± 0.15 |
> | 10 | 0.92 |4.37 ± 0.30 |
> | 15 | 0.86 |3.70 ± 0.40 |
>
>
> >(Q3: Sensitivity of IDEAS to the inital population quality):
>
> We evaluate the sensitivity of IDEAS to the quality of the initial population using two analyses.
> (1) As shown in our response to W2, IDEAS is robust to noisy labels: even with high noise ($\eta=10$), AUC drops by only 8%.
> (2) Through analysis on the TF dataset, we vary the training set size ($N_T$) and observe that reducing $N_T$ from 47.5K to 50 results in only a 6% drop in AUC. This robustness is attributed to the use of COLOR for attribution, which has relatively few trainable parameters (Table 2).
>
> | $N_T$   | AUC (mean ± std) |
> |-|-|
> | 47487 | 5.83 ± 0.16  |
> | 1000  | 5.69 ± 0.24  |
> | 500   | 5.44 ± 0.21   |
> | 100   | 5.60 ± 0.28   |
> | 50    | 5.49 ± 0.36   |
>
> [1] Pandey et al., ACS JCIM, 2025.
>
> [2] Dallago et al., NeurIPS, 2021.

---

> > ### Author Rebuttal · Reviewer_HbLk · 2026-04-03
> >
> > Thanks for your response. I will maintain my positive score.

---

### Official Review · Reviewer_TQjm · 2026-03-13

**Soundness:** 3
**Presentation:** 3
**Significance:** 2
**Originality:** 2
**Overall Recommendation:** 4
**Confidence:** 3

**Summary:**

This paper presents IDEAS (Interpretable Evolutionary Approach for biological Sequences), an active-learning style framework for biological sequence design under tight oracle budgets. At each iteration, IDEAS selects top-performing sequences, trains an explainable surrogate model to predict the property of interest, and computes position-wise attributions. These attributions are then used to perform interpretable mutations: the method preferentially mutates low-attribution regions and replaces them using high-attribution motifs extracted from top sequences. A temperature-controlled sampling strategy supports both exploitative and exploratory mutations. The authors evaluate IDEAS across six sequence design tasks with multiple oracle budgets, comparing against a diverse set of baselines spanning evolutionary, RL-based, and generative approaches. The paper reports consistent improvements in sample efficiency (via an AUC-based metric) and analyzes trade-offs between optimization performance, diversity, and novelty, alongside ablations studying the impact of key components (e.g., temperature parameters, mutation strategy variants, and different attribution methods).

**Compliance With Llm Reviewing Policy:**

Affirmed.

**Final Justification:**

Most of my initial concerns are properly addressed by the authors during the rebuttal period. I therefore raise my score to weak accept.

**Key Questions For Authors:**

1. How does IDEAS perform when the attribution signal is noisy or partially unfaithful (e.g., smaller surrogate capacity, limited training data, deliberately corrupted attributions, or stochastic/noisy oracle feedback)?
2. What is the wall-clock runtime and training overhead per iteration (training the explainable surrogate), and how does it compare to the baselines under the same oracle budget?
3. The paper notes that increasing motif size can improve performance (at least initially). What is the rationale for fixing m=1 across all datasets? Is there a dataset-dependent recommended choice, and how sensitive are results to m?
4. Relative to which set are novelty and diversity computed (initial dataset vs. all evaluated sequences so far)? If novelty is computed relative to the evolving pool, does it bias against later iterations?

**Limitations:**

Yes.

**Strengths And Weaknesses:**

# Strengths:
- Method is technically coherent and well-specified: the iterative loop is logically consistent and easy to follow.
- Empirical evaluation is fairly extensive: multiple tasks, multiple budgets, multiple baselines, repeated trials, and ablations provide reasonable evidence.
- Ablations provide useful insight into which components matter (e.g., temperature parameters; comparisons against an exploit-only variant), which strengthens the causal story.

# Weaknesses:
- The approach relies on a key assumption: attribution scores are sufficiently faithful and stable. While the paper argues that “faithful” attribution methods work similarly, it remains unclear how robust IDEAS is when attributions are imperfect—arguably the more realistic regime in hard biological design settings (noisy oracles, limited data, model misspecification).
- A notable portion of the performance gain appears to come from exploitative, attribution-guided local moves, while the exploratory component contributes mainly to diversity/novelty rather than the primary AUC metric. This raises the question of whether IDEAS is primarily a sophisticated guided hill-climbing heuristic, and whether its advantages persist when local optimization is insufficient.
- The paper would benefit from a clearer accounting of computational cost (e.g., wall-clock time and training overhead of retraining the explainable surrogate each iteration).

---

> ### Author Rebuttal · Authors · 2026-03-30
>
> Thank you for your thoughtful review. We appreciate your recognition of IDEAS' technical coherence, extensive empirical evaluation, and the ablation studies that establish causal understanding of key design choices. Below, we address your questions and comments. All new experimental results discussed below will be incorporated into the revised paper.
>
> ## Weaknesses (W) and Questions (Q)
> >(W1, Q1: IDEAS performance under noisy attribution score and limited training data):
>
> We evaluate sensitivity to attribution quality $\phi$ by injecting noise into training labels:
> $$Y_{\text{noisy}} = Y + \mathcal{N}(0, (\sigma(Y)\eta)^2),$$
> where $\eta$ controls noise level and $\phi$ quality is measured via Pearson correlation (PCC) with ground truth (GRAVY index). The results demonstrate two key robustness properties: **(1) robust to attribution noise:** a 66% drop in PCC ($\eta=0 \rightarrow 10$) leads to only an 8% reduction in AUC; **(2) robust to noisy training data:** even at large noise ($\eta=10$), AUC remains above the second-best baseline (AdaLead, AUC = 26.9). Performance degrades to AdaLead's level only when $\phi$ is effectively random ($\eta=50$).
>
> |$\eta$|PCC|AUC|
> |-|-|-|
> |0|0.99|37.10±1.97|
> |0.5|0.97|36.77±1.71|
> |1|0.91|37.04±2.00|
> |2|0.87|35.73±2.55|
> |5|0.76|36.42±2.68|
> |10|0.60|34.22±2.29|
> |50|-0.1|26.44±4.79|
>
> **Effect of training dataset size ($N_T$):** On the TF dataset, reducing $N_T$ from 47.5K to 50 samples results in only a **6% drop in AUC**, attributed to COLOR's lightweight architecture (Table 2), making IDEAS practical in data-scarce settings.
>
> |$N_T$|AUC|
> |-|-|
> |47K|5.83 ± 0.16|
> |1000|5.69 ± 0.24|
> |500|5.44 ± 0.21|
> |100|5.60 ± 0.28|
> |50|5.49 ± 0.36|
>
>
> >(W3, Q2: Wall-clock time comparison):
>
> Wall-clock time is measured on the GRAVY index dataset, where the oracle is an analytical expression. As shown below, IDEAS sits between lightweight evolutionary methods (AdaLead, CMA-ES) and deep generative models (Cbas, Dbas), while being 16× and 50× faster than GFN-AL and DynaPPO, respectively. All the models are run on an AMD EPYC 7413 24-Core Processor for this study.
>
> |Method|BO|CMA-ES|AdaLead|Cbas|Dbas|GFN-AL|DynaPPO|IDEAS|
> |-|-|-|-|-|-|-|-|-|
> |Runtime (s)|6.2|0.033|0.003|2.8|3.1|21|65|1.3|
>
>
> >(W2: Is IDEAS a sophisticated hill-climbing algorithm?):
>
>
> While IDEAS mutates the top-$B$ sequences at each iteration, it is fundamentally distinct from hill-climbing: mutations are guided by motif-level attribution scores (Equations 4--9) aggregated across the entire top-$B$ population, not applied randomly. This population-level signal reflects the collective fitness landscape, promoting diversity and preventing premature convergence. Consequently, IDEAS achieves on an average **25% higher sample efficiency and 55% higher diversity than AdaLead**, demonstrating that attribution-guided mutation overcomes the limitations of hill-climbing.
>
>
> >(Q3: Rationale behind choosing the motif size):
>
> **Rationale for selecting motif size**
>
> We select the motif size ($m$) based on the COLOR [1] model, which partitions a sequence into motifs of length $m$ and learns a global representation by aggregating their latent embeddings. In practice, $m$ is chosen to maximize predictive performance (e.g., $R^2$, MAE, MSE) during COLOR training, and we use this optimal $m$ in IDEAS.
>
> **Is $m=1$ always recommended?**
>
> No. On two new datasets (AAV and Thermostability; see W1/Q1, Reviewer Xuze), IDEAS achieves 21% higher AUC than the best baseline at $m{=}10$, while reducing to $m{=}1$ drops AUC by 12% on average, confirming that larger motifs are necessary for longer sequences. The choice of $m{=}10$ follows COLOR's predictive performance, as shown in the table below (AAV dataset), where both predictive performance and AUC peak at $m{=}10$.
>
> |m|Predictive Performance (PCC)|Mean AUC|
> |-|-|-|
> |1|0.86|3.90|
> |3|0.87|3.92|
> |6|0.89|3.97|
> |10|0.92|4.37|
> |15|0.86|3.70|
>
>
> >(Q4:Relative to which dataset, novelty and diversity is calculated):
>
> For fair comparison of diversity and novelty, we standardize evaluation across methods using a common fitness threshold $y_c$.
>
> - After all design iterations, we select a shared target fitness value $y_c$.
> - For each method, we identify the iteration $t$ where the mean fitness $\langle y_t \rangle$ of generated sequences reaches $y_c$, where $y_t \in \mathbb{R}^{B \times 1}$ denotes the fitness values of the $B$ generated sequences at iteration $t$, and denote the corresponding sequences as $x_t \in \mathbb{R}^{B \times L}$.
>
> We then compute:
> - **Diversity**: Mean pairwise edit distance among sequences in $x_t$.
> - **Novelty**: Mean edit distance between $x_t$ and the initial population $x_0$ (common across methods).
>
> Using a shared $y_c$ ensures comparisons are made at equivalent fitness levels, enabling fair evaluation.
>
> [1] Pandey et al., ACS JCIM, 2025.
>
> [2] Beck et al., Journal of Structural Biology, 1998.
>
> [3] Dallago et al., NeurIPS, 2021.

---

> > ### Author Rebuttal · Reviewer_TQjm · 2026-04-03
> >
> > I thank the authors for their detailed responses. Most of my concerns are properly addressed. I will raise my score accordingly.

---

### Decision · Program_Chairs · 2026-04-30

**Decision:**

Accept (regular)

**Comment:**

The reviewers agreed that the underlying idea is well-founded, novel, and thoroughly evaluated. The paper clearly demonstrates strong empirical performance and is clear and well-written. During the discussion, the authors addressed concerns about dependence on the XAI attribution quality and expanded to more challenging design tasks, buttressing the most important weaknesses of the original submission.